# Evaluation of Groundwater Using an Integrated Approach of Entropy Weight and Stochastic Simulation: A Case Study in East Region of Beijing

**DOI:** 10.3390/ijerph18147703

**Published:** 2021-07-20

**Authors:** Yongxiang Zhang, Ruitao Jia, Jin Wu, Huaqing Wang, Zhuoran Luo

**Affiliations:** 1Faculty of Architecture, Civil and Transportation Engineering, Beijing University of Technology, Beijing 100124, China; yxzhang@bjut.edu.cn (Y.Z.); Jiaruitao@emails.bjut.edu.cn (R.J.); wang@univ-lehavre.fr (H.W.); zrluo@emails.bjut.edu.cn (Z.L.); 2LOMC, UMR CNRS 6294, Université du Havre, 76600 Le Havre, France

**Keywords:** groundwater quality, IWQI, hydrochemistry, sensitivity analysis, Chaoyang district

## Abstract

Groundwater is an important source of water in Beijing. Hydrochemical composition and water quality are the key factors to determine the availability of groundwater. Therefore, an improved integrated weight water quality index approach (IWQI) combining the entropy weight method and the stochastic simulation method is proposed. Through systematic investigation of groundwater chemical composition in different periods, using a hydrogeochemical diagram, multivariate statistics and spatial interpolation analysis, the spatial evolution characteristics and genetic mechanism of groundwater chemistry are discussed. The results show that the groundwater in the study area is weakly alkaline and low mineralized water. The south part of the study area showed higher concentrations of total dissolved solids, total hardness and ${\mathrm{NO}}_{3}^{-}$-N in the dry season and wet season, and the main hydrochemical types are ${\mathrm{HCO}}_{3}^{-}$-Ca and ${\mathrm{HCO}}_{3}^{-}$-Ca-Mg. The natural source mechanism of the groundwater chemical components in Chaoyang District includes rock weathering, dissolution and cation exchange, while the human-made sources are mainly residents and industrial activities. Improved IWQI evaluation results indicate that water quality decreases from southwest to northeast along groundwater flow path. The water quality index (WQI) method cannot reflect the trend of groundwater. Sensitivity analysis indicated that the improved IWQI method could describe the overall water quality reliably, accurately and stably.

## 1. Introduction

Groundwater plays an important role in social production, living and ecological maintenance [1,2,3,4]. With the acceleration of urbanization and the rapid growth of population, the world’s dependence on groundwater resources is further deepening. In addition to the amount of groundwater resources, groundwater quality is also an important factor to determine the availability of groundwater resources [5,6,7,8]. In most cities in the world, attention to groundwater has exceeded their groundwater quantity and concerns regarding groundwater quality, as water quality has implications for water safety, and research into groundwater quality can provide useful information for sustainable management of water resources. Therefore, safe and hygienic groundwater resources are a primary focus of related research [9,10,11]. Therefore, in order to alleviate the shortage of water resources and the deterioration of groundwater quality, safe and hygienic groundwater resource is a primary focus of related research [12,13].

Beijing is a world-class metropolis with a large population and developed economy, which has a great demand for water resources. For a long time, urban life and economic development have been heavily dependent on groundwater. In order to understand the hydrochemical characteristics of groundwater and the suitability of various uses (drinking, irrigation, and industrial activities), many researchers have conducted groundwater quality assessment of Beijing [14,15,16]. Past studies have rarely carried out studies on groundwater in the Chaoyang District, and up to now, there is no deep understanding of the spatial evolution characteristics and genetic mechanisms. This study systematically assesses groundwater hydrochemistry and water quality throughout the Chaoyang District.

In order to understand the hydrochemical characteristics, genesis and environmental significance of groundwater in Chaoyang District, a Piper diagram [17], Gibbs diagram [18], ion ratio [19,20] and self-organizing water quality index method were used to analyze the groundwater samples in the wet season and dry season in 2019. Traditional water quality methods, for example, the individual parameter method, were replaced by the water quality index (WQI), which was firstly developed by Horton [21]. The WQI method can transform a large amount of water quality data into a single number representing the water quality status of the study area [13,22]. Despite the fact that the WQI method is widely used in groundwater quality assessment, its drawbacks such as subjective weights, inflexible structure and inadequate input parameters are obvious, and the efforts to develop improved WQI have been continued [23,24]. The traditional WQI method is mainly controlled by water quality parameters and parameter weights, so reasonable weights and parameters are very important for the accuracy of evaluation results [13,25].

The main purposes of the current study were to: (1) analyze the hydrochemical characteristics, ion sources and hydrochemical formation of groundwater; (2) study the spatiotemporal characteristics and evolution driven environmental factors of groundwater chemistry; (3) use the improved IWQI (integrated weight water quality index ) method to evaluate groundwater quality. Therefore, this study determined 14 water quality parameters through the literature collection, and used the stochastic simulation method to calculate the objective weight. Then, the entropy weight method is used to calculate the subjective weight to avoid subjectivity [13,26,27,28]. Finally, the comprehensive weight is determined by the self-organization water quality evaluation method to evaluate the suitability of groundwater [29]. It is expected to provide theoretical reference for the exploitation and utilization of groundwater resources and ecological environment protection in the Chaoyang District.

## 2. Study Area

### 2.1. Location and Climate

The study area (39°49′ to 40°05′ N, 116°21′ to 116°38′ E) is located in the Beijing Plain (Figure 1b). The district’s north–south length is 28 km, and east–west width is 17 km, leading to a total area of 470.8 km^2^ (Figure 1c). The population in the Chaoyang District is approximately 3.47 million in 2019. It has a temperate continental and semi-arid monsoon climate with an annual average temperature of 11.6 °C, the frost-free period is 192 days, and it has 2841.4 h of annual daylight. The average annual evaporation is 1200 mm/year exceeding the average annual precipitation (581 mm/year), and summer precipitation accounts for 75% of the whole year.

### 2.2. Geology and Hydrogeology

The Chaoyang District is located between the northeast of the Beijing depression and the northern of Daxing District uplift. So, Cenozoic stratigraphic deposits are primarily controlled by these two tectonic structure units, and its basement is mostly consisted of Neocathaysian in NE direction. The Quaternary sediments cover the Paleozoic strata with a maximum thickness of 450m, and are composed primarily of sands, silty clay, and gravels (Figure 1d).

The study of area topographic elevation varies from 20 to 45 m, with a slope of 1/100~1/2500. The overall topography is gentle and gradually decreases from northwest to southeast. The main aquifer consists of Quaternary loose sediments. The maximum Quaternary thickness is located in the east-central area. The groundwater level of the Chaoyang District varies between 4 and 40 m in 2019, with an average of 24.7 m [30]. The vast majority of Chaoyang District is construction land (Figure 1c).

## 3. Materials and Methods

### 3.1. Sample Collection and Analysis

Groundwater samples were collected and tested from 28 monitoring wells in April (dry season) and September (wet season) in 2019. Before sampling, pumping the accumulated water from monitoring wells for 5–10 min until the flowing water presents a stable temperature, pH value, dissolved oxygen and Eh value [23,24]. The collection, processing, storage and analysis of samples follow the standard procedures recommended by the Ministry of Water Resources of China [31].

Through the literature search, 14 water quality parameters such as total dissolved solids (TDS), total hardness (TH), major anions (${\mathrm{HCO}}_{3}^{-}$, Cl^−^, ${\mathrm{SO}}_{4}^{2-}$), major cations (K^+^, Na^+^, Ca^2+^, Mg^2+^) and other minor elements (Fe^3+^, F^−^, NO_3_-N, NO_2_-N, pH) were determined and analyzed. Among those chemical parameters, a multiparameter portable measuring instrument was applied to measured pH in suit; the contents of K^+^, Na^+^, Ca^2+^, Mg^2+^, Fe^3+^ were analyzed using inductively coupled plasma atomic emission spectrometry (ICP-AES). Total hardness (TH) was measured by the Na_2_EDTA titrimetric method. An electric blast-drying oven, along with an electronic analytical balance (vapor-drying method), was used to measure TDS; ${\mathrm{HCO}}_{3}^{-}$, Cl^−^, ${\mathrm{SO}}_{4}^{2-}$, F^−^, ${\mathrm{NO}}_{3}^{-}$ and ${\mathrm{NO}}_{2}^{-}$ were detected by water quality-inorganic anion determination-ion chromatography [32,33,34,35]. The accuracy of water quality detection was controlled by blank samples and parallel samples, and only charge balance errors less than ±5% of water samples were accepted.

### 3.2. Data Analysis

The improved IWQI method was utilised to assess the groundwater quality for drinking. Further, data analysis was carried out using various software. SPSS (v16.0, Inc., Armonk, NY, USA) and ArcGIS (v10.6, ESRI, Redlands, CA, USA) were employed for chemometric analysis and spatial map preparation, respectively. Hydrochemical facies and water types were identified through the Piper diagram using AqQA software (version 2010.1, Waterloo Hydrolgeologic, Kitchner, ON, Canada). A Gibbs diagram was used to analyze the chemical components of groundwater.

### 3.3. Self-Organization Assessment Method

In this study, the groundwater of the Chaoyang District was evaluated by the self-organization water quality assessment method. It can make the weight of water quality parameters more reasonable, solve the problem of the evaluation results being too subjective or objective, and make it more in line with the actual situation [36,37]. It includes four steps: calculation of subjective weight by the entropy-weighted quality index (EWQI), calculation of the objective weight by the stochastic simulation approach (SSA), calculation of the integrated weight by Equations (6)–(8) and water quality assessment based on the IWQI.

#### 3.3.1. Calculation of Subjective Weight through EWQI

Entropy was borrowed from thermodynamics by Shannon [26], which objectively reflects the useful information. The specific steps are as follows:

The first step is constructed an evaluation matrix. The original data evaluation matrix of the corresponding evaluation index is shown in Equation (1):(1)$$R={\left[\begin{array}{cccc} r_{11} & r_{12} & \ldots & r_{1n}\\ r_{21} & r_{22} & \ldots & r_{2n}\\ \ldots & \ldots & \ldots & \ldots \\ r_{m1} & r_{m2} & \ldots & r_{mn} \end{array}\right]}_{m\times n}$$
where *m* (*i* = 1, 2, 3, …, *m*) represents the number of water samples and *n (j* = 1, 2, 3, …, *n*) represents the number of evaluated parameters.

The specific gravity of the parameter value *P_ij_* of the *i*-th water sample and *j*-th evaluated parameter is calculated as shown in Equation (2).
(2)$$P_{ij}=\frac{r_{ij}}{\sum_{i=1}^{m}r_{ij}}\left(i=1,2,3,\ldots m;\;j=1,2,3,\ldots ,n\right),$$

Then, entropy value (*e_j_*) and entropy weight (subjective weight: *W_sj_*) were calculated as follows:(3)$$e_{j}=-\frac{1}{\ln m}\sum_{i=1}^{m}P_{ij}\ln P_{ij}\left(1,2,3,\ldots ,m\right),$$
(4)$$W_{sj}=\frac{1-e_{j}}{\sum_{j=1}^{n}\left(1-e_{j}\right)}\;\left(j=1,2,3,\ldots ,n\right),$$

#### 3.3.2. Weight Based on Stochastic Simulation Approach (SSA)

The stochastic simulation approach (SSA) was used as an objective weighting method in the present study to calculate the relative weights of variables, avoiding the short comings of conventional information entropy which ignores the importance of the index itself [29]. In this study, a database of water quality parameter weights obtained from the published literature was used. The relative weights for water quality parameters from different literature sources are given in Table 1. The weight of parameters collected from different literature was taken as the database of water quality parameters, and the mean value and standard deviation of water quality parameters in the database were calculated. Then, based on the database of water quality parameter weights, a large dataset of random weights (2000 simulations) was generated by the stochastic simulation approach (SSA). SSA uses weights for water quality parameters to aggregate those and find the final value. The values of objective weight (*W_oj_*) are calculated as:(5)$$W_{oj=}\;Norminv\left(v,m,s\right),$$

The *Norminv* function is the inverse of the normal probability distribution function, where v is the function generator rand () conforming to the uniform distribution of [0, 1], *m* is the mean value of parameters in the dataset of random weights, and *s* is the standard deviation of parameters in the dataset of random weights.

#### 3.3.3. Calculation of Integrated-Weight

The integrated weight (*W_j_*) can be expressed as follows:(6)$$w_{j}=W_{sj}\times W_{oj}/\sum_{j}^{n}W_{sj}\times W_{oj},$$
(7)$$G=\sum_{j=1}^{n}[{\left(w_{j}-W_{sj}\right)}^{2}+\left(w_{j}-W_{oj}{)}^{2}\right],$$
(8)$$W_{j}=GW_{sj}+\left(1-G\right)W_{oj},$$
where *G* is the preference coefficient and *G* ∈ [0, 1].

#### 3.3.4. Calculation of Self-Organizing Water Quality Assessment

After calculating the weights, determination of the quantitative rating scale *Q_j_* of parameter *j* using Equation (9), the *IWQI* for each sample is calculated by Equation (10)
(9)$$\{\begin{array}{c} Q_{j}=100\times (C_{j}-C_{jp})/\left(T_{j}-C_{jp}\right)\\ Q_{pH}=100\times \left(C_{jpH}-7\right)/\left(8.5-7\right) \end{array},$$
(10)$$IWQI=\sum_{j=1}^{m}W_{j}Q_{j},$$
where *C_j_* is the concentration of each parameter *j* (mg/L), *C_jp_* is the ideal value of the parameter in pure water (consider *C_jp_* = 0 for all, except pH where *C_jp_* = 7), and *T_j_* is the standard value for each chemical indicator according to the Chinese Quality Standard for Groundwater (mg/L).

According to the self-organizing water quality index method, groundwater is divided into five categories, namely excellent, good, medium, poor and extremely poor [55,56,57]. The evaluation result based on the IWQI method is shown in Table 2.

#### 3.3.5. Sensitivity Analysis

The sensitivity analysis in this paper is mainly to determine the rationality of the selected groundwater index and analyze the accuracy of the evaluation results. In general, the higher the sensitivity value, the more significant the impact of indicators on the stability of evaluation results [58]. The calculation formula is as follows Equation (11).
(11)$$S_{i}=\frac{\left|V_{i}/N-v_{i}/n\right|}{N_{i}}\times 100\%$$
where *S_i_* is the sensitivity value calculated after removing the *i*-th index, *V_i_* is the IWQI score of *i*-th groundwater sample, *v_i_* is the IWQI score with *i*-th index removed, and *N* and *n* are the number of parameters for calculating *V_i_* and *v_i_*, respectively.

## 4. Results and Discussion

### 4.1. Descriptive Statistics of Chemical Components in Groundwater

According to the analysis results of the main ion concentration of groundwater in the Chaoyang District of Beijing (Table 3), including minimum, maximum, mean and % of sample exceeding standard (% of SES), the groundwater presents a weak alkaline environment. The average pH values in the wet season and dry season were 7.69 and 7.54, respectively, and the standard deviation of pH value in the wet season and dry season is 0.33 and 0.25, respectively, indicating that the pH value in the groundwater is relatively stable. TDS values were 248–1040 mg/L and 165–993 mg/L in the dry season and wet season, respectively, with average values of 527.75 mg/L and 495.29 mg/L. TH reflects the lithologic characteristics of the stratum in groundwater, the concentration of TH in the dry season and wet season varied from 88 to 711 mg/L, 70–633mg/L, and the mean values are 373.64 and 344.32 mg/L, and 35.71% and 32.14% of samples exceed acceptable limits (450 mg/L), respectively. From the macro component’s point of view, the average concentration of main cations in groundwater in the dry season from large to small is Ca^2+^ > Na^+^ > Mg^2+^ > K^+^, the average concentration is 87.84 mg/L, 59.58 mg/L, 37.48 mg/L, 1.95 mg/L, respectively, the average concentration of main anions from large to small is ${\mathrm{HCO}}_{3}^{-}$ > ${\mathrm{SO}}_{4}^{2-}$ > Cl^−^, and the average concentration is 327.89 mg/L, 79.88 mg/L, 76.17 mg/L, respectively. The relationship of ion concentration in the wet season is consistent with that in the dry season. However, the average concentration of ions in the dry season was slightly higher than that in the wet season [59]. Among the macro ions, only ${\mathrm{HCO}}_{3}^{-}$ exceeded the limit of 250 mg/L WHO (2011) [60], and the exceeding rate was 82.14%. The mean concentration of F^−^ was 0.32 mg/L in the dry season and 0.34 mg/L in the wet season, which did not exceed the permissible limit. Fe^3+^ is an essential element for human beings, but in the dry season, 14.29% of the samples exceeded the limit (0.30 mg/L) required by WHO (2011) [60]. The average concentrations NO_2_-N and NO_3_-N in the wet season were 0.0022 and 5.16 mg/L, respectively. The mean concentrations NO_2_-N and NO_3_-N in the dry season were 0.0003 and 6.36 mg/L, respectively. In general, the concentrations of NO_2_-N and NO_3_-N were at a low level in the wet season and dry season. Only 3.57% of the groundwater exceeded the standard NO_3_-N (>20 mg/L) in the dry season and wet season.

### 4.2. Spatial Characteristics of TDS, TH and ${NO}_{3}^{-}$-N in Groundwater

In order to further analyze the composition and spatial distribution of groundwater in the study area, the Kriging method of spatial interpolation in the Arcgis 10.6 software was used to analyze TDS, TH and NO_3_-N. The application of the spatial interpolations approach to the class of major ions indicates that these ions have the same spatial characteristics [61]. However, the distribution area of each index in the dry season can basically cover the distribution area of each index in the wet season. This may be caused by the climatic characteristics of the study area. After fresh water diluted by precipitation enters the groundwater [62], there is a decrease in hydrochemical composition [63].

As shown in Figure 2a,b, the concentration of TDS in groundwater mainly ranged from 60 mg/L to 650 mg/L, and the high concentration of TDS (exceed 1000 mg/L) in the groundwater during the dry season is mainly distributed in the south of the study area. However, the high concentration of TDS is only distributed in the southwest of the study area. It can also be seen from Figure 2c,d that the concentration of TH in groundwater mainly varies from 60 mg/L to 650 mg/L in different periods, and the spatial distribution of TH is consistent with that of TDS. The high concentration of TH (exceed 650 mg/L) is mainly distributed in the south of the study area. The concentration of NO_3_-N was mainly between 6 mg/L and 23 mg/L in the wet season and dry season (Figure 2e,f). The high concentration area of NO3-N (exceed 20 mg/L) was mainly distributed in the southwest of the study area. It can be seen that the high concentration of main chemical components in groundwater is mainly distributed in the southwest of the study area approaching the metropolitan area of Beijing. Therefore, the spatial distribution of chemical components in the Chaoyang District is closely related to the number of residents and industrial production.

### 4.3. Hydrochemical Facies

The Piper diagram [17] can reflect the chemical composition of groundwater, and then can identify the hydrochemical characteristics of groundwater and its controlling factors [31]. The difference in ion concentration between the wet season and dry season is small. As shown in Figure 3, in the triangle composed of cations in the lower left corner, the groundwater sample points are mainly located in area B, and the cations are mainly Ca^2+^. In the triangle diagram of the anion composition, the samples are mainly concentrated in the E region, and the dominant anions are ${\mathrm{HCO}}_{3}^{-}$ and ${\mathrm{CO}}_{3}^{2-}$. In the diamond diagram, the sample points are mainly concentrated in area 1, indicating that the hydrochemical types of groundwater in the study area are mainly ${\mathrm{HCO}}_{3}^{-}$-Ca or ${\mathrm{HCO}}_{3}^{-}$-Ca-Mg. Very few groundwater samples fall in zone 3 of the diamond diagram. As exhibited in the figure, the hydrochemical type is relatively simple, which is mainly dominated by the HCO_3_-Ca·Mg-type water. The analyses show a seasonal difference which is not significant in the main groundwater chemical contents constraining the chemical characteristics changes only slightly, and the hydrochemical type remains unchanged [32].

### 4.4. Source analysis of Main Hydrochemical Components

#### 4.4.1. Natural Control Factor

A Gibbs diagram is mainly used in the study of groundwater chemical components. It reflects the control factors of main ions in groundwater macroscopically and qualitatively judges the source of water chemical composition [3,18]. According to the relationship between TDS, Na+/(Na^+^ + Ca^2+^) and Cl^−^/(Cl^−^ + ${\mathrm{HCO}}_{3}^{-}$), the natural origin of main chemical components in water can be divided into precipitation, water–rock interaction and evaporation concentration. The Cl^−^/(Cl^−^ + ${\mathrm{HCO}}_{3}^{-}$) ratio of most groundwater samples is less than 0.5 (Figure 4b), indicating that the hydrochemical composition of these groundwater samples is affected by water–rock interaction such as rock weathering. With the increase of Cl^−^/(Cl^−^ + ${\mathrm{HCO}}_{3}^{-}$) ratio, TDS value also showed a gradually increasing trend, indicating that the chemical composition of groundwater is affected by other factors. According to the relationship between TDS and Na+/(Na^+^ + Ca^2+^), there is no obvious change in the ratio of Na+/(Na^+^ + Ca^2+^) in groundwater, indicating that the influence of additional factors on the composition of main cations in groundwater is not significant.

#### 4.4.2. Analysis of Main Ion Sources in Groundwater

The use of ionic ratios in groundwater can be used to reveal the main factors controlling the hydrochemistry of geochemical processes. In general, the dissolution of rock salt is the main source of Na^+^ and Cl^−^ in groundwater, and it releases equal amounts of Na^+^ and Cl^−^. Figure 5a shows the scatter plot of Na^+^ against Cl^−^, in which all the groundwater samples are plotted on the upper and lower sides of the *y* = *x* relationship line [19], indicating that the dissolution of rock salt is not the primary hydrogeochemical process affecting the hydrochemistry components of groundwater. As shown in Figure 5b, the groundwater samples were distributed near or below the *y* = *x* relationship line, reflecting that weathering of carbonate and silicate rocks was the main factor of geochemical processes in the study region. Figure 5c shows that although the process of carbonate weathering has an important effect on chemical characteristics of groundwater in study area, other Ca^2+^ may exist, such as Ca^2+^ containing silicates or cation exchange.

The groundwater samples mainly fall in the upper sides of the *y* = *x* relationship line in Figure 5d, indicating that Ca^2+^ and ${\mathrm{SO}}_{4}^{2-}$ in the groundwater of study area did not mainly originate from the dissolution of gypsum. The ratio of Ca^2+^/Mg^2+^ can be used to analyze the influence of carbonate rock and silicate weathering on groundwater hydrochemical characteristics [37]. As shown in Figure 5e, the groundwater samples were mainly plotted on between the *y* = *x* and *y* = 0.5*x* relationship lines, indicating that the weathering of carbonate rock is the main rock weathering process affecting the hydrochemical characteristics of groundwater of the study area. Generally speaking, cation exchange may be one of the important processes affecting the chemical characteristics of groundwater in a region [9,64]. In addition, the comparison diagram of (Ca^2+^ + Mg^2+^) − (${\mathrm{HCO}}_{3}^{-}$ + ${\mathrm{SO}}_{4}^{2-}$) and (Na^+^ − Cl^−^) was used to further illustrate the ion exchange between groundwater and its host environment. As shown in Figure 5f, the slope fitting line was close to −1, which indicates that cation exchange is an important process affecting the water chemical characteristics of groundwater in the study area.

### 4.5. The Comparison between IWQI and WQI in Groundwater

Based on 14 groundwater quality indexes and WHO (2011) [60], the groundwater quality in the study area was evaluated by self-organizing water quality evaluation method. The results of the water quality assessment in the study area are shown in Table 2. The IWQI values in the dry season are between 29.364 and 106.53, with an average of 57.65, based on the IWQI value, the classification results of groundwater samples show that 0.00%, 46.43%, 35.71%, 14.29% and 3.57% are excellent, good, medium, poor and very poor categories, respectively. The IWQI value of the wet season is between 20.95 and 100.81, with an average of 54.94. Based on the IWQI value, the classification results of groundwater samples show that 3.57%, 46.43%, 32.14%, 14.29% and 3.57% belong to the excellent, good, medium, poor and extremely poor categories, respectively. The WQI values in the dry season are between 30.671 and 89.107. The classification results of groundwater samples show that 0.00%, 39.29%, 35.71%, 53.57% and 7.14% are excellent, good, medium, poor and very poor, respectively. The WQI values in the wet season are between 23.244 and 74.998. The classification results of groundwater samples show that 3.57%, 50.00%, 46.43%, 0.00% and 0.00% are excellent, good, medium, poor and very poor categories, respectively.

As shown in Figure 6a,b, the spatial distribution of evaluation values in the dry season are basically the same as the wet season, and the IWQI value decreases from southwest to northeast of the study area. The IWQI value exceeded 75 (poor, extremely poor), which was distributed in the south of the study area. IWQI values lower than 75 (excellent, good and medium) were mainly distributed in the northeast of the study region. As shown in Figure 6c,d, the spatial evaluation values in the dry season are different from the wet season. In the dry season, the WQI value exceeded 75 but did not exceed 100 (poor), which was distributed in the southwest of the study area. In the wet season, the WQI value does not exceed 75, and no poor or extreme levels of water appear. The results indicate that WQI method drawbacks such as subjective weights and inflexible structures are obvious, and the improved IWQI method is more accurate for groundwater quality.

### 4.6. Sensitivity Analysis of Assessment Index

The influence of removing each input index, in addition to TDS and the sensitivity of each index in the wet and dry season, was less than 1%, which ranged between 0.093–2.442% and 0.034–2.417% in Figure 7, respectively. It can be seen that the most positive effect on the IWQI scores is related to removing TDS in the dry season and wet season, which were 2.442% and 2.417%, respectively. In the dry season, the influence of Cl^−^ on the evaluation results is the least, with an average sensitivity of 0.093%. However, during the wet season, the average sensitivity of pH is the lowest, which is 0.034%. Sensitivity analysis showed that the influence of any one parameter on IWQI results was not strong. In brief, all indexes play a role in water quality assessment, and the IWQI does not only rely on one or a few parameters, which is consistent with other researches [64]. The results of sensitivity analysis show that the self-organizing water quality assessment method was developed correctly.

## 5. Conclusions

The different ion concentrations, hydrochemistry, and spatial interpolation of 28 groundwater samples in the Chaoyang District, east of Beijing, were investigated and analyzed. The groundwater quality for drinking purposes was assessed using an integrated approach of entropy weight and stochastic simulation methods. The following conclusions were reached:

The concentration of TDS, TH and NO_3_ are dominant in the groundwater, and the spatial distribution is higher in the southwest than in the northeast. Under the joint influence of natural geological conditions and human activities, hydrochemical types of groundwater in Chaoyang District are mainly ${\mathrm{HCO}}_{3}^{-}$-Ca and ${\mathrm{HCO}}_{3}^{-}$-Ca-Mg types, and there is little difference between the chemical types of groundwater in wet season and dry season. The chemical ions in groundwater in the study area are mainly controlled by rock weathering. Ca^2+^ and Mg^2+^ come from the dissolution of carbonate, silicate and evaporite, and Na^+^ from the dissolution of rock salt, among which the dissolution of carbonate plays a dominant role. The main driving factors of groundwater chemical evolution are carbonate weathering, water-karst filtration and human activities.

The improved IWQI method was applied to groundwater quality evaluation. The evaluation results show that the groundwater levels in the study area are various, with the good level being the most, followed by the medium level, and the excellent and poor level being the least. The medium and poor level groundwater quality are primarily located in the high groundwater table area around the southwest of the study area. It may be caused by shallow groundwater depth, and it experiences strong evaporation.

The results of the sensitivity analysis show that the improved IWQI method did not rely too much on one or several specific groundwater parameters, and it can be used as an effective method to evaluate groundwater quality. The methodology based on this study will be useful for local water resource managers for developing strategies to mitigate and prevent groundwater contamination.

## Figures and Tables

**Figure 1 ijerph-18-07703-f001:**
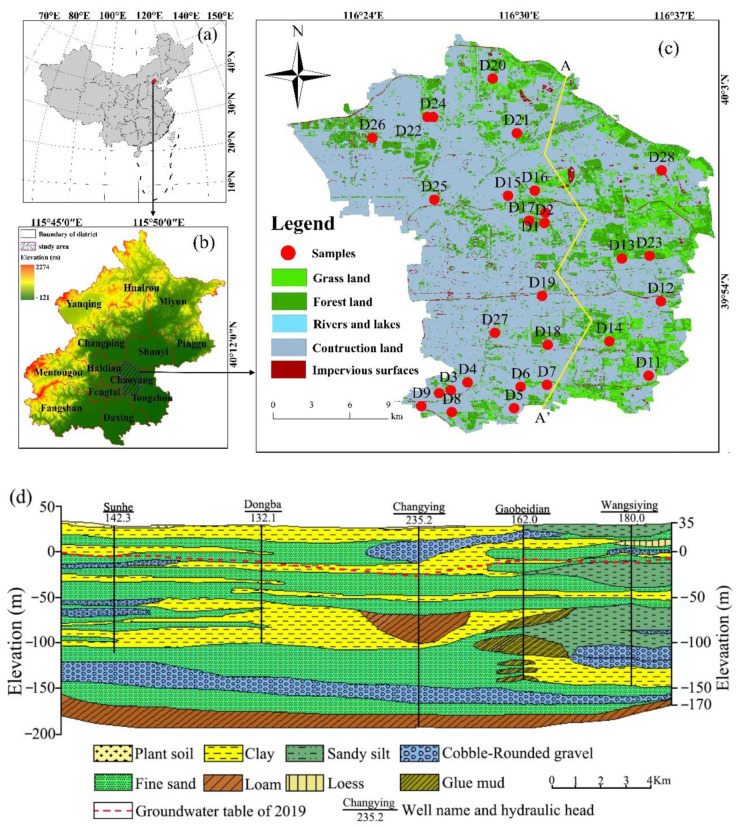
Maps showing (**a**) the location of Beijing in China. (**b**) the location of study area in Beijing. (**c**) the sampling locations of groundwater in study area combined with land use. (**d**) A-A’ hydrogeological profile in the study area.

**Figure 2 ijerph-18-07703-f002:**
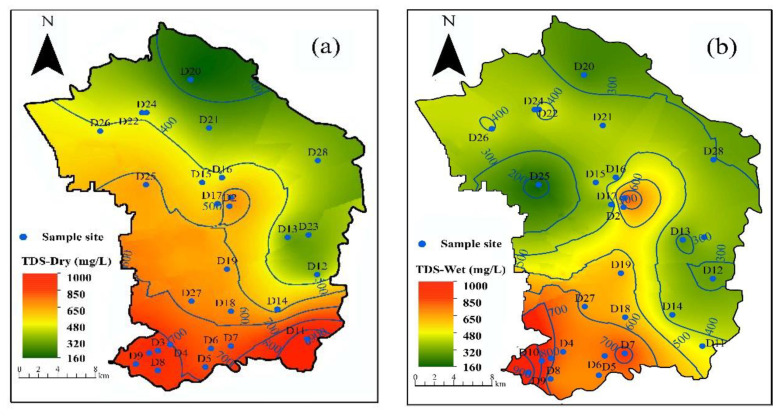
Spatial distribution of groundwater quality in the study area. (**a**) TDS in dry season. (**b**) TDS in wet season. (**c**) TH in dry season. (**d**) TH in dry season. (**e**) NO^3^-N in dry season. (**f**) NO^3^-N in wet season.

**Figure 3 ijerph-18-07703-f003:**
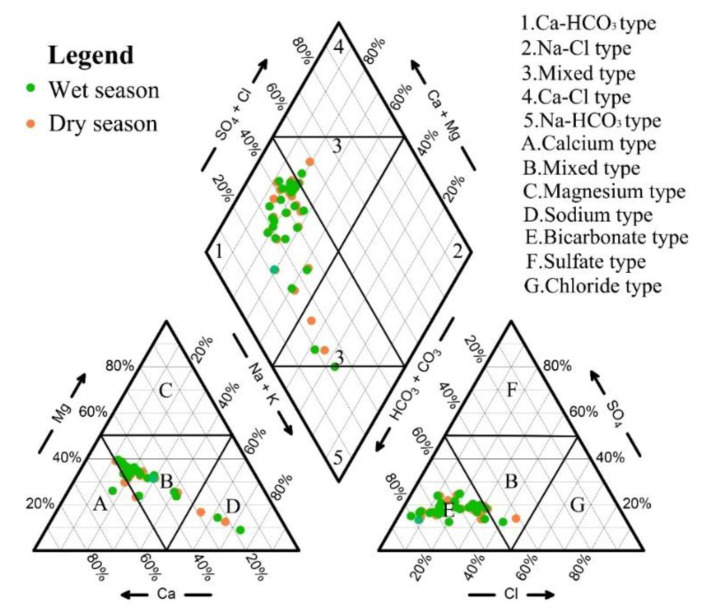
The Piper diagram of groundwater in the study area.

**Figure 4 ijerph-18-07703-f004:**
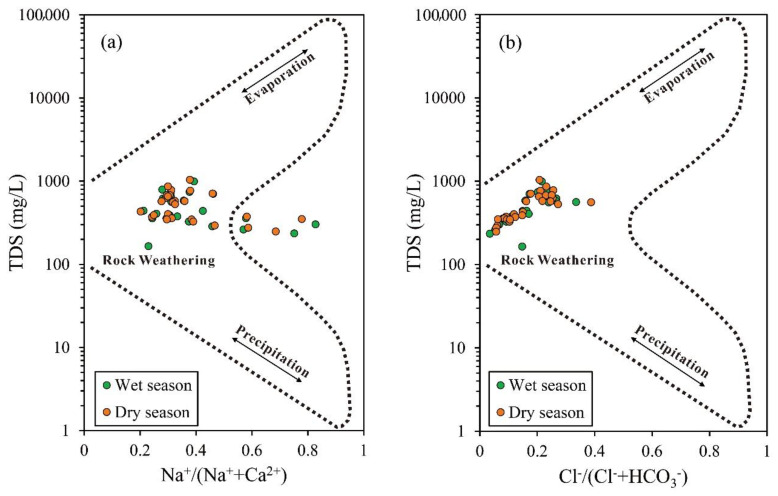
Gibbs analysis chart of groundwater quality in the study area. (**a**) [Na^+^/(Na^+^ + Ca^2+^)] vs. TDS. (**b**) [Cl^−^/(Cl^−^ + ${\mathrm{HCO}}_{3}^{-}$)] vs. TDS.

**Figure 5 ijerph-18-07703-f005:**
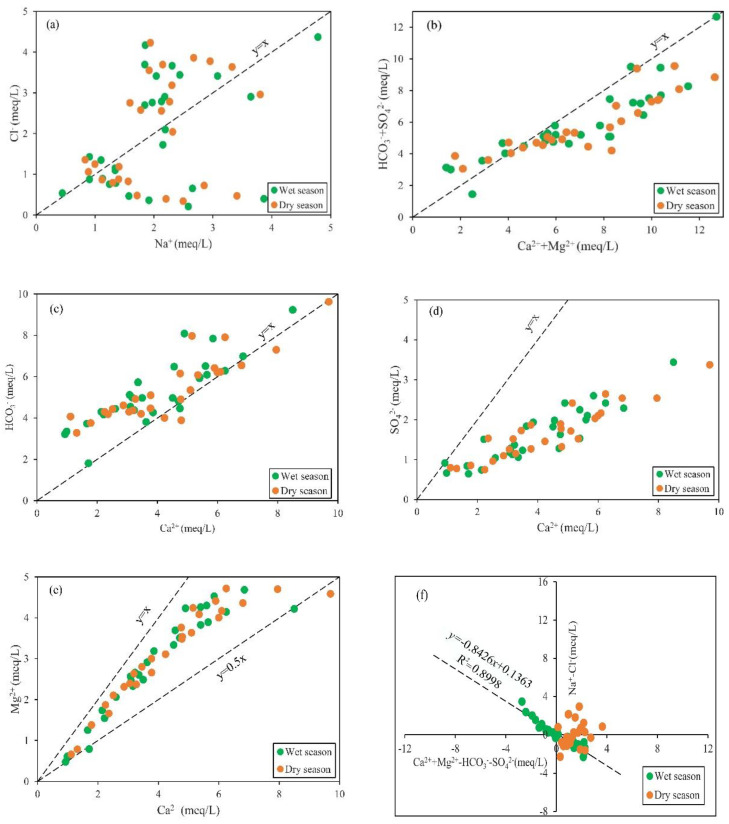
Ion ratio diagram of groundwater in study area. (**a**) Na^+^ vs. Cl^−^. (**b**) [Ca^2+^ + Mg^2+^] vs. [${\mathrm{HCO}}_{3}^{-}$ + ${\mathrm{SO}}_{4}^{2-}$]. (**c**) Ca^2+^ vs. ${\mathrm{HCO}}_{3}^{-}$ (**d**) Ca^2+^ vs. ${\mathrm{SO}}_{4}^{2-}$. (**e**) Ca^2+^ vs. Mg^2+^. (**f**) [Ca^2+^ + Mg^2+^ − ${\mathrm{HCO}}_{3}^{-}$ − ${\mathrm{SO}}_{4}^{2-}$] vs. [Na^+^ − Cl^−^].

**Figure 6 ijerph-18-07703-f006:**
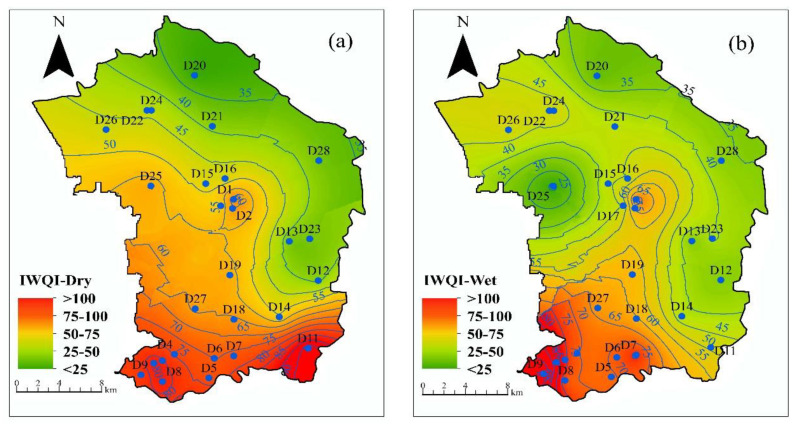
Distribution map of IWQI and WQI in the study area. (**a**) IWQI value in dry season. (**b**) IWQI value in wet season. (**c**) WQI value in dry season. (**d**) WQI value in wet season.

**Figure 7 ijerph-18-07703-f007:**
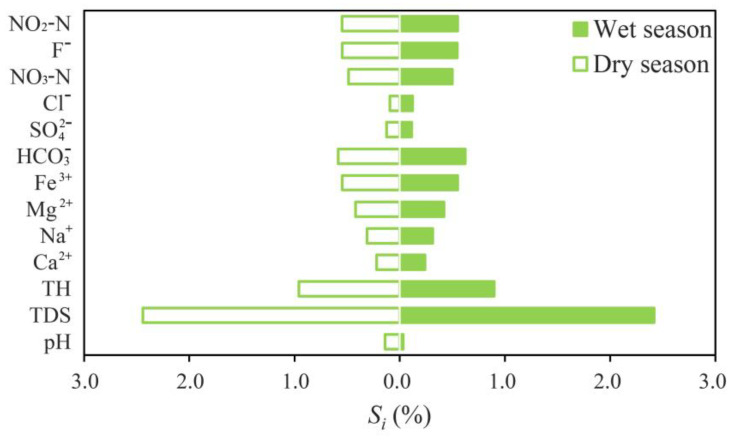
Sensitivity analysis of different water quality parameters based on IWQI method.

**Table 1 ijerph-18-07703-t001:** The weights of water quality evaluation parameters were collected through different literature.

pH	TDS	TH	Ca^2+^	Na^+^	Mg^2+^	K^+^	Fe^3+^	${\mathrm{HCO}}_{3}^{-}$	${\mathrm{SO}}_{4}^{2-}$	Cl^−^	NO_3_-N	F^−^	NO_2_-N	R
0.013	0.038	0.056	0.016	0.023	0.016			0.015	0.021	0.027	0.136	0.375	0.184	[37]
0.083	0.104	0.042	0.042	0.083	0.042	0.021	0.083	0.021	0.104	0.104	0.083	0.083		[38]
0.032	0.063	0.063	0.048	0.063	0.048		0.079	0.016	0.048	0.063	0.079		0.079	[39]
0.041	0.068	0.041					0.041		0.068	0.068	0.068	0.068	0.054	[40]
0.069	0.069	0.023					0.023		0.069	0.068	0.114		0.114	[41]
0.070	0.040	0.138	0.076	0.053	0.127			0.191	0.058	0.063	0.038	0.098	0.024	[37]
0.093	0.080	0.096		0.120			0.134		0.098	0.082	0.375	0.121	0.360	[42]
0.143	0.176	0.071	0.071	0.107	0.071	0.071		0.071	0.107	0.107				[43]
0.080	0.110	0.060	0.060	0.080	0.060			0.020	0.080	0.110	0.110	0.110		[44]
0.103	0.064		0.026	0.077	0.026	0.026		0.077	0.051	0.077		0.128		[45]
0.071	0.071	0.071	0.048	0.071	0.048	0.048		0.048	0.095	0.095	0.119	0.119		[11]
0.050	0.050		0.020	0.040	0.020	0.020		0.020	0.070	0.100				[44]
0.114	0.114		0.057	0.086	0.057	0.057		0.029		0.114	0.143			[9]
0.093	0.116	0.070		0.047	0.023	0.047		0.070	0.093	0.070	0.116	0.116		[46]
0.103	0.172		0.069	0.103	0.069	0.034		0.069	0.103	0.103	0.172			[47]
0.068	0.114	0.045	0.068	0.114	0.068	0.045		0.023	0.114	0.114	0.114	0.114		[48]
0.110			0.060	0.060	0.060	0.030		0.080	0.110	0.080	0.140	0.110		[49]
0.031	0.031	0.031	0.031	0.063	0.031	0.031	0.094	0.031	0.031	0.031	0.063	0.063		[50]
0.070		0.023		0.023					0.070	0.070	0.116		0.116	[51]
0.069		0.063	0.063	0.070			0.084			0.070	0.070	0.126		[52]
0.094			0.094	0.063	0.063	0.063	0.125	0.063	0.125	0.094				[53]
0.135	0.108	0.054	0.054		0.027		0.135		0.108	0.081	0.135	0.108		[54]

Note: TDS, total dissolved solids. TH, total hardness. R, References.

**Table 2 ijerph-18-07703-t002:** Evaluation results of self-organizing water quality evaluation method at different season.

Season	IWQI Rank	<25	25–50	50–75	75–100	>100
	Water Quality	Excellent (Ⅰ)	Good (Ⅱ)	Medium (Ⅲ)	Poor (Ⅳ)	Extremely Poor (Ⅴ)
Dry season	No. of sample	0	13	10	4	1
	Percentage (%)	0.00	46.43	35.71	14.29	3.57
Wet season	No. of sample	1	13	9	4	1
	Percentage (%)	3.57	46.43	32.14	14.29	3.57

Note: IWQI Rank, Integrated water quality index Rank.

**Table 3 ijerph-18-07703-t003:** Statistical analysis of chemical composition of groundwater.

Indexes	Dry Season					Wet Season				
	Min	Max	Mean	SD	% of SES	Min	Max	Mean	SD	% of SES
pH	7.01	8.13	7.57	0.25	0.00	7.04	8.62	7.69	0.33	3.57
TDS	248	1040	527.75	196.03	3.57	165	993	495.29	194.41	0.00
TH	88	711	373.64	153.45	35.71	70	633	344.32	147.37	32.14
Ca^2+^	22.3	194	87.84	39.44	0.00	18.6	170	80.07	35.88	0.00
Na^+^	19.1	118	49.58	21.69	0.00	10.2	110	46.65	21.62	0.00
Mg^2+^	7.89	56.6	37.48	13.98	0.00	5.74	56.2	35.04	14.58	0.00
K^+^	0.46	3.34	1.95	0.63	0.00	0.44	3.11	1.86	0.56	0.00
Fe^3+^	0.02	16	0.19	0.41	14.29	0.01	0.15	0.02	0.03	0.00
${\mathrm{HCO}}_{3}^{-}$	200	587	327.89	90.83	82.14	110	563	318.86	96.26	82.14
SO_4_^2−^	35.9	162	79.88	30.58	0.00	30.9	165	77.41	31.91	0.00
Cl^−^	12.1	154	76.17	47.64	0.00	7.34	155	69.67	46.44	0.00
NO_3_-N	0.258	22.6	6.36	6.21	3.57	0.018	20.3	5.16	5.77	3.57
F^−^	0.135	0.753	0.32	0.10	0.00	0.087	0.795	0.34	0.12	0.00
NO_2_-N	0.0003	0.0003	0.0003	0.00	0.00	0.0003	0.0142	0.0022	0.00	0.00

Note: The unit of water quality parameters are mg/L, except pH, SD, standard deviation, % of SES, % of sample exceeding standard.

## Data Availability

Data available on request due to privacy and ethical restrictions.

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
