# Peer review of "Evaluation of Groundwater Using an Integrated Approach of Entropy Weight and Stochastic Simulation: A Case Study in East Region of Beijing"

_ijerph, 2021, doi:10.3390/ijerph18147703_

Round 1
Reviewer 1 Report
This manuscript presents a study on groundwater evaluation by using a new approach based on entropy weights and stochastic simulation. I think the manuscript is concise and well-structured: methodology and results are quite clear. I also think that this manuscript will be of interest to the readers of this journal and should be published after some minor/moderate revisions.
Moderate comments:
- The writing style needs to be greatly improved. Two examples:
Line 36 “it is the key to realize…”
Line 38 “economic”, do you mean “economy”?
- Since you introduce a new method (IWQI), I would like to see its comparison to the IWQ values. For example, you could add two plots corresponding with the IWQ dry and IWQ wet in Figure 6 (or in figure 2/new figure). This way you can add more discussion in section 4.
- Lines 59-64 “Therefore, this study…..”, I suggest moving these lines after paragraph 65-72. This way, the manuscript explains first the goals of the study and afterward how it is done.
- Nomenclature needs to be revised. In its current state, it is confusing:
R in equation (1) should be bold
p in equation (3) is not explained. Is p=P? If so, pick one.
Also, later in Equation (7) appears another P.
In equation (4), Wsj is the same wsj in equation (6)?
Line 156, is Cj or cj?
Both, Equation (9) and equation (11) use the term S for different things.
Minors:
- In the abstract, what does IWQI stands for?
- Lines 136-137. Could you explain the shortcomings?
- Table 1. I would add two rows to include the mean “m” and standard deviation “s” for each parameter.
- The reference WHO (2011) (e.g. in line 195) is missing from the references.
Reviewer 2 Report
Dear Authors, thank you for the opportunity of reading your paper. It is interesting. I can see you put a lot of effort in writing. Literature list is very rich (63 positions), up to date, you really put your attention to a references review. I like the quality and clearlity of your figures. Although I have some suggestions to improve your paper:
- Why did you choose 2019 as your research epoch, can you give me some explanation? Is it representative year?
- Is taking only one dry season and only one wet season give enaught conclusions? Did you try to add more data?
- I am not sure if you can write TEMPORAL in figure 2, it is only one year, you have rather spatial instead of spatial and temporal distribution.
- In paragraph 4.1 you wrote "The average ph value in dry season is 7.57, and that in wet season is 7.69. The ph value in dry indicating that ph value in groundwater is relativly stable" .How do you know that when you check only one dry season. Moreover, you wrote this in a very deterministic way, but you did not compare it with a time series.
- Are those value 7.57 and 7.69 in paragraph 4.1 are for averaged wells?
- Figure 2 - do you present here an average or a sum?
- Table 1 - here you made a big work, looking for a simmilar rsearch. But I really miss an interpretation and evaluation od data in this huge table. Can you also refer to your research.
Reviewer 3 Report
Comments on the paper “ Evaluation of groundwater using an integrated approach of entropy weight and stochastic simulation: a case study in east region of Beijing” by Zhang et al.
The paper describe an interesting approach for the investigation of water quality in a Chinese watershed, analyzing major ions and main water parameters. While the paper is generally interesting and the methods seems to be rigorous, there are different major issues which hampers a clear understanding of the main results.
Firstly, the English language and the quality of presentation need to be improved. The paper, in fact, presents different unclear or too generic sentences, which make it unclear and unnecessarily long. Some examples are in lines 32- 36, in lines 62-66, in lines 177- 178, in line 181 and line 185, the whole section 4.4.2, lines 335-336, but there are different others along the text.
Moreover, the main aim of the paper is not clear in my opinion. In fact, If the main focus is the methodology, this feature needs to be better highlighted in the whole paper. As it is, from the title and the materials and methods it seems that Self-organization assessment method and entropy weight play a major role in the paper results, but then this part is only shortly discussed in the final part of the results. Therefore, since it seems that other approaches are helpful for the investigation of water quality, authors should then move the focus on them, e.g., spatial interpolations and seasonal trend analysis (see e.g. Soltani et al. 2020 https://doi.org/10.1007/s10661-020-08572-z , or Binda et al. 2020 https://doi.org/10.1007/s10653-019-00405-4 ).
Moreover, there are different minor issues which require a revision in my opinion:
-abstract, lines 17-18: please add a sentence with the main drawings from this result, and not just list them;
-lines 83-85: It should be good to add the described tectonic structures in the map of Figure 1;
-figure 1: Please indicate the spatial location of the section in the map. I suggest to use a bigger font size for the point labels in Figure 1b to make them more readable and revise the misspelling in Figure 1d (gravel instead of gravle).
-line 108: How do the authors analyzed TDS and TH thorugh ICP-AES? Please indicate the standard method used.
-line 127: I think the authors should define what is an evaluation value, since this kind of modeling is not often applied in this field.
-Line 195: Which are the WHO limits for bicarbonates? Please report them in the text.
-lines 206-208, 231-232 and 245-247: Please move in the method section.
-Figure 2: Please indicate the panels and to which parameters they refer in caption. Moreover, panel b) present a different colro scale compared with other panels.
-Lines 239-241: This paragraph is too generic. Are these samples behaving to the same springs? Are these related with the tectonic structures? Please discuss in more detail this feature.
-Lines 302-306: This issue needs to be discussed in accordance with the section 4.2. The same spatial trend was interpreted in a completely different way in that section, considering the urban context and the possible anthropic effects.
-Line 322: Please reintroduce the main aim of the study in the first sentence of the conclusion section.
Round 2
Reviewer 3 Report
After the revisions made by the authors, the quality of the manuscript and the clarity has clearly improved. There still are few minor issues to be revised (mostly typos) which are listed below:
- line 23: "cannot" instead of "can't"
- line 117: "Analysis" instead of "nanlysis"
- line 118: "to assess" instead of "to assessment"
- line 119 :"Various softwares:" instead of "various software"
- line 122: "was used to analyze" instead of "is used analysis"
- lines 145-146: "avoiding the short comings of conventional information entropy which ignores" instead of "which avoided the short-145 comings of conventional information entropy ignores"
- line 282: "Fig. 5c and shows". Unclear, please rewrite
- line 288: "in Fig. 5d" instead of "(Fig. 5d)"
- Line 314: The with capital L
- Lines 320 and 325: do the authors mean "spatial distribution" with the term "spatial"?
Author Response
Please see the attachment.

This manuscript is a resubmission of an earlier submission. The following is a list of the peer review reports and author responses from that submission.